

# The significance of soil properties to the estimation of soil moisture from C-band synthetic aperture radar

John Beale[1], Boris Snapir[1], Toby Waine[1], Jonathan Evans[2], and Ronald Corstanje[1]

[1]School of Water, Energy and Environment, Cranfield University, Cranfield MK43 0AL, UK
[2]Centre for Ecology & Hydrology, Wallingford OX10 8BB, UK

**Correspondence:** John Beale (john.e.beale@cranfield.ac.uk)

**Abstract.** Soil Moisture is a key variable in hydrology, weather and climate modelling. Research has been directed to the estimation of soil moisture over wide areas through a combination of modelling, in-situ measurement and remote sensing to improve the accuracy of hydrological and meteorological forecasting. For monitoring and controlling irrigation and other agricultural purposes, there is also a need to capture local variability. Significant soil moisture differences are observed between and within fields due to land use, soil properties, drainage, tillage, vegetation, solar radiation, air temperature, wind, rain and other factors. Taking the United Kingdom as an example, the average area of agricultural fields is about 12 hectares, requiring a mapping resolution of less than 100m. Satellite-based remote sensing, including the use of C-band SAR (such as on Sentinel-1), has the potential to satisfy this requirement, but many current data products are aggregated to a spatial resolution of at least 1km and/or provide soil moisture in relative units or indices. Both strategies mitigate the uncertainties introduced by field-scale variability in soil hydrological and vegetation properties. Geospatial datasets of soil properties and land use, crop modelling and other remote sensing techniques may provide an alternative approach to mitigating this variability and allow finer scale products to be produced with acceptable errors. This paper looks at the role of soil properties in the estimation of soil moisture from C-band SAR. We show that information on the soil texture, organic matter content, surface temperature, land use and crop modelling should be important inputs to the success of retrieving soil moisture at the field scale. Previously published data provides guidance in setting soil roughness parameters, based on soil properties, following farming operations such as primary tillage. Beyond soil moisture retrieval, there is exciting potential in SAR remote sensing data to improve the spatial resolution and mapping accuracy of some soil properties.

## 1 Introduction

Soil moisture (SM) is one of the main driving parameters in the engineering, agronomic, ecological, biological and hydrological functions of the soil. Knowledge of SM, readily available water (RAW), or drought indices (Pablos et al., 2017) is valuable in agriculture to maximise crop yield whilst minimising irrigation to preserve water resources (Adeyemi et al., 2017; Molden and James, 2000). Assessment of flood risk is improved by knowledge of the capacity of the soil to absorb rainfall (Alvarez-Garreton et al., 2014; Brocca et al., 2017). The ability of the soil to exchange water and heat with the atmosphere has a strong effect on local weather, so soil moisture is an important, but often unknown, input to meteorological models (Dorigo and





de Jeu, 2016; Entekhabi et al., 1996). SM varies significantly in depth, spatial and temporal extent (Brocca et al., 2017) as it responds to soil properties, drainage, vegetation, land management, solar radiation, air temperature, wind, rain and other factors. Significant differences are observed between and within fields. Reliable and timely maps, at field scale, of surface soil moisture or metrics related to the field capacity, saturation and permanent wilting point of the soil (Chandler et al., 2017) would

be especially useful to farmers and growers. It is already well established that SM has to be interpreted in the context of depth, soil texture and structure for most of these applications.

In situ soil moisture measurement techniques include soil moisture probes, and cosmic ray soil moisture sensors (CRS). The latter are based on neutron SM sensors (Andreasen et al., 2017) (installed above ground) that exploit the elastic scattering (slowing) of fast neutrons from cosmic rays by hydrogen nuclei (protons) in water molecules. This reduces the number of fast

neutrons above the soil as they become slow, or thermal, neutrons. The fast neutron count can be processed to measure the average soil moisture in a volume of soil contained in a radius of approximately 200m (Köhli et al., 2015) and a depth which is moisture dependent. A cosmic-ray soil moisture observing system (COSMOS) has been set up in the UK (Evans et al., 2016), USA (Zreda et al., 2012), Australia (Hawdon et al., 2014) and Germany (Zacharias et al., 2011). These are among several examples of extensive ground measurement networks, but they still represent only a sparse sample of the different conditions

across their regions. They are valuable for research purposes, including the validation of remote sensing soil moisture products, and may have a role to play in assimilation of data with remote sensing and/or land surface models, such as the Joint UK Land Environment Simulator (JULES) (Clark et al., 2010).

Soil moisture estimation by satellite remote sensing offers the potential for wide area coverage at minimal cost to the user. There are several options. Optical multispectral imaging (Periasamy and Shanmugam, 2017) exploits the dependence

of the optical reflectance of bare soil on moisture content, and thermal infrared (Hain et al., 2009) techniques exploit the thermal inertia of soil due to soil moisture. Both are confounded by vegetation obscuring the soil, cloud cover, and certain lighting conditions. Passive microwave sensors exploit the relationship between the microwave emission of bare soil and its moisture content. They are characterised by a low spatial resolution (Drinkwater et al., 2009; Petropoulos et al., 2015; European Space Agency, 2017), but are relatively unaffected by illumination or clouds, with L-band sensors being minimally affected

by vegetation. Examples of the latter are SMOS (50km resolution) (Kerr et al., 2001) and the passive element of SMAP (40km resolution) (Entekhabi et al., 2010). To achieve higher spatial resolution with microwave systems requires active radar, which exploits the relationship between the radar backscatter coefficient and soil moisture (Oh et al., 1992). Non-imaging radar systems include scatterometers (Bartalis et al., 2007) and radar altimeters (Uebbing et al., 2017). Synthetic aperture radar (SAR) is an active microwave imaging technology with high spatial resolution (10-50m). Several L, C and X band

SAR satellites have been launched offering the spatial resolution and frequency of coverage that most closely addresses the requirement of near real-time field-scale moisture mapping. This study focuses on C-band because data from the European Space Agency's Sentinel-1 satellites is freely available, enabling regular derivation of soil moisture products at the required scale and resolution.

The physics that relates radar backscatter to soil moisture is very complex with many unknown factors. Within the large

volume of recently published research on this topic, a key priority is to eliminate, mitigate or quantify these unknowns, usually





by remote sensing means alone. However, for decades, soil scientists have been collecting information on soil properties such as texture, density, organic matter content and hydrological properties, and assembling this data into detailed maps. Detailed maps of land use are also available. The quality and spatial resolution of these maps are continuously improving, assisted, not least, by the use of remote sensing techniques. How can this valuable resource be usefully applied to the soil moisture

estimation problem? Few studies discuss the impact of soil properties on SAR soil moisture estimation, with some authors reporting a negligible effect, although this is acknowledged by some (Baghdadi et al., 2007) as a problem with the current resolution and accuracy of soil maps and databases.

The objective of this work is to examine and quantify the potential influence of soil properties, in their broadest sense, on the retrieval and interpretation of soil moisture from SAR data, with a focus on C-band. The impact of soil properties on each of

the significant factors that affect the radar backscatter will be examined, identifying and quantifying the uncertainties that soil properties can introduce. Finally, the significance of this in terms of future algorithm development and further research will be summarised.

In this paper, we will refer to soil moisture in the volumetric unit, $m_v$, which is the ratio of water volume within the total soil volume, expressed as a percentage.

## 15  2  C-band SAR soil moisture retrieval

The interest in remote sensing of soil moisture (SM) from space is driven by its coverage, scalability and the availability of high resolution synthetic aperture radar (SAR) data, such as the C-band instrument on ESA's Sentinel-1 (S1) satellites (Doubkova et al., 2016). The starting point for soil moisture retrieval is usually the SAR backscatter coefficient $\left(\sigma^0\right)$ which is easily obtained or processed from online satellite data services. $\sigma^0$ is a function of many variables associated with the

viewing geometry and land surface, of which soil moisture is only one. Some of these factors are known (such as the SAR configuration), but rigorous retrieval of soil moisture would require independent observations of the other factors. Where these are not readily available, estimation approaches have been developed as a pragmatic solution. However, significant errors will be introduced and the data product will have limited validity.

### 2.1  Factors affecting C-band SAR soil moisture estimation

The process of estimating soil moisture from C-band SAR data is complicated by scattering from vegetation (Lang and Sidhu, 1983), scattering due to soil surface roughness (Álvarez-Mozos et al., 2009; Martinez-Agirre et al., 2017), temperature dependence (Rodionova, 2017b), scattering from man-made objects (Tadono et al., 2000) and sources of radio frequency interference (RFI) (Monti-Guarnieri et al., 2017).

In the absence of reliable data to quantify these effects, assumptions and approximations are made, particularly with regards

to soil texture, organic matter, soil temperature and vegetation cover over the site or area of interest. The processing also has to take account of incidence angle (Wang et al., 2017) and the reduction of noise in the image due to speckle. To validate estimates against ground measurements, regression analysis is often employed, where the metrics used to assess the capability



of the SAR estimation process are the bias (the systematic offset) and the coefficient of determination, $R^2$ (capturing the impact random variation due to uncertainties and noise). An ideal process that always provides an accurate agreement with ground measurements would have a bias of zero and $R^2$ value of $1.0$ . Estimation of soil moisture using such a process may be subject to a wide confidence interval and limited to certain spatial or temporal contexts (van der Velde et al., 2012),

land uses and vegetation types. A study of the accuracy of SM estimation from SAR in Australia (Doubková et al., 2012) confirms that surface roughness, terrain and vegetation can have a significant impact and can result in no significant correlation between SAR derived SM estimates and estimates derived from observations ($^2 < 0.22$) over significant areas of the country. Understanding and quantifying the contributory factors by independent measurements, modelling or ancillary data sets, should enable improved algorithms to be developed to give greater precision and a better understanding of the accuracy of the estimate.

We first discuss the problem of vegetation before considering the contribution of soil to these confounding factors.

### 2.1.1   Vegetation

Vegetation is often the most significant factor to account for in SAR soil moisture estimation as layers of vegetation may cover all of the soil surface, attenuating both the transmitted radar signal and the backscatter from the underlying soil. Additionally, there is a contribution to the backscatter coefficient due to scattering within the vegetation layers. Ulaby et al. (1984) reported

that, for typical agricultural crops, where the Leaf Area Index (LAI) is above approximately 0.5 the vegetation contribution driving radar backscatter dominates over the soil properties (moisture and roughness). As crops develop and the LAI exceeds 0.5 the retrieval of soil moisture retrieval by C-band SAR will be challenging and perhaps impossible. Where there is potential to remove vegetation effects, the tool frequently used to estimate $\sigma_{soil}^0$, is the Water Cloud Model (WCM) (Attema and Ulaby, 1978; Graham and Harris, 2003); it is an empirical model that relates a metric of vegetation activity to the SAR backscatter

and transmission properties of the vegetation layer. There are many examples of the use of remotely sensed indices, such as LAI and Normalised Difference Vegetation Index (NDVI), being used as the metric; but these may not be suitable in wetter climates where cloud-free observations of these indices from multispectral satellites (such as Landsat-8 or Sentinel-2) may be very infrequent. Other options are biomass estimates obtained from SAR (Kumar et al., 2018; Pichierri et al., 2018), exploiting the temporal and polarisation characteristics of volume scattering in vegetation. The WCM has empirical parameters

that describe other sources of variability, which may include land use, vegetation type, and clutter sources. Optimum values of the parameters are determined that minimise the errors in the model prediction (based on in situ ground measurements) compared to the observed radar backscatter. The limitations of the in situ data, with regard to being spatially representative and measurement depth, affect the scope and validity of the model.

For continuous tree canopies, the Michigan Microwave Canopy Scattering (MIMICS) model (Ulaby et al., 1988) was devel-

oped, followed by MIMICS II (McDonald and Ulaby, 1991), which models broken tree canopies. MIMICS requires parameters including the volume fractions of leaves, branches and trunks within the canopy, LAI and the total water density and biomass of each of the canopy constituents. The MIMICS model is often used in preference to WCM for some vertically stemmed crops, such as wheat (Ulaby et al., 1990) and maize (Monsivais-Huertero and Judge, 2011). Whilst, the use of such a model





requires detailed land use information, including accurate knowledge of the crops and their stage of development, it has been shown to be effective in some studies (Liu et al., 2017a, b).

### 2.1.2 Surface Roughness

The radar backscatter of bare soil is a combination of that due to diffuse scattering from within the soil (which is a function of soil moisture) and that due to diffuse scattering from the soil-air interface (which is a function of its roughness, not soil moisture). With the wavelength of a C-band radar at 5.4 GHz being about 5.5 cm, comparable to the size of the surface features, scattering from the soil-air interface is significant and must be accounted for when estimating the soil relative permittivity (dielectric constant), $\varepsilon_{soil}$ from its radar backscatter coefficient, $\sigma^0_{soil}$. This is further complicated by cross and along track differences in the soil surface relief that is often introduced by tillage and the passage of machinery. There are a number of empirical and semi-empirical models that are used for this purpose, including the Advanced Integrated Equation Model (AIEM) (Fung et al., 1992), and models proposed by Oh et al. (1992) and Dubois et al. (1995). The role of these models is to quantify the impact of surface scattering due to the roughness and periodicity of the soil surface. This is normally described in the remote sensing community by the roughness parameters $H_{rms}$ ("rms height") (Bryant et al., 2007) and $L_c$ (correlation length), or a parameter which combines both, symbolised as $Z_g$ (Zribi et al., 2014). $H_{rms}$ is equivalent in approximate magnitude to the Random Roughness (RR) (Currence and Lovely, 1970) parameter often used by agriculturalists and soil conservationists, which is the standard deviation of the soil height relative to a plane of best fit through the soil surface. The determination of these parameters by remote sensing is of interest, because surface roughness is one of the main sources of $m_v$ retrieval errors from satellite SAR sensors (Martinez-Agirre et al., 2017; Lievens et al., 2009). A number of strategies to eliminate surface roughness effects in the context of soil moisture retrieval from SAR were reviewed by McNairn and Brisco (2004), who discovered that to separate the effects of soil moisture and soil surface roughness requires diversity in measurement (Gorrab et al., 2016) comprising, multi-frequency (such as combining different SAR sensors (Zhang et al., 2018)), multi-angle (which might be achieved with different orbits of the same sensor (Wang et al., 2016)), or multi-polarisation. Mattia et al. (1997) found that a co-polarised correlation coefficient based on circular polarisation is strongly correlated with surface roughness, but not soil moisture. Unfortunately, fully polarimetry data is often not available, Sentinel-1 only provides one co-polarised (VV) and one cross-polarised (VH) channel. Simultaneous retrieval of soil roughness and soil moisture is not satisfactory when one radar configuration is used (Baghdadi et al., 2018b), without making significant assumptions.

### 2.1.3 Surface Temperature

In most situations, soil temperature has a minor influence on the C-band dielectric properties of soil (Jackson, 1987) though the effect is observable in Sentinel-1 data (Rodionova, 2017b). Generally the effect is ignored as insignificant compared to variability introduced by the other factors discussed. This assumption breaks down when the temperature is low enough for some or all of the soil water to freeze.

When soil begins to freeze it is characterised by a sharp decrease in the real part of the relative permittivity ($\varepsilon'_r$) , as documented by Zhang et al. (2003), Mironov and Lukin (2009) and Mironov et al. (2017b). This is easily detectable in Sentinel-





imagery and can be used as a method for mapping of frozen soil (Baghdadi et al., 2018a; Park et al., 2011). The problem

for soil moisture estimation is that frozen soil will appear to have a much lower soil moisture content than it actually has. In

the case of overnight and early morning frost there may be patches of frozen and unfrozen soil within a field. Correction for

this effect might involve masking the frozen areas (using a method after Baghdadi et al. (2018a)), or selective application of a

correction factor to the frozen soil pixels.

Another problem in some areas is dew; water that condenses (and wets the surface) when the ground temperature falls

below the dew point of the air immediately above it. Long wave infrared emission causes the ground surface to cool at night,

especially on clear nights with little wind. Warmer air can hold more moisture, and most soils can supply water for evaporation

and transpiration, increasing the absolute humidity of the air (higher dew point), making dew formation more likely, as the air

only has to cool relatively little before crossing the dew point temperature. The presence of dew as a layer on the vegetation

(and perhaps a temporary wetting of the soil surface) gives rise to another source of error in soil moisture estimation. The

impact of dew on SAR backscatter was extensively studied by the TerraDew project (Riedel et al., 2002a, b), where formation

of dew was found to cause a 1-2dB reduction in $\sigma^0$, with the cross-polarised band being affected more than the co-polarised

band.

### 2.1.4  Local Incidence Angle

The radar backscatter coefficient, $\sigma^0$, is dependent on the local incidence angle (LIA) between the incoming SAR beam from

the satellite and the normal to the local ground surface. An excellent discussion of the topic is given by van Zyl et al. (1993).

There two calibration issues; the angular dependence of the radar antenna pattern, and the projected area of the pixel, that are

usually accounted for in higher level satellite data products. A third issue is the angular dependence of scattering from soil, its

roughness and vegetation, as described by models such as the IEM (Shi et al., 1995) and MIMICS (Ulaby et al., 1988). In many

other soil moisture retrieval approaches, such as change detection, it is common to remove this dependency by normalising $\sigma^0$

empirically to one LIA value by methods such as histogram equalisation (Mladenova et al., 2013) or linear regression (Widhalm

et al., 2018). The LIA for each pixel may be calculated from a digital elevation model (DEM) and the satellite position. This is

also a source of potential error as there are errors and holes in DEM data (Wechsler, 2007). DEM's are rarely updated, so any

change in the land surface topography due to erosion, earthquake and landslip may not be represented in the DEM.

### 2.2  Methods for Soil Moisture Retrieval from SAR

To understand the context in which soil properties may be important, this section will briefly discuss the main approaches for

soil moisture estimation from SAR. The use of SAR data to provide higher temporal and spatial resolution when combined

with low resolution passive microwave data is a further use of SAR that is excluded from this study.

A combination of axiomatic and empirical models have been developed to predict microwave emissions and the SAR

backscatter coefficient, $\sigma^0_{soil}$, of the land surface. If the other inputs can be quantified, soil moisture may be estimated from

measured $\sigma^0$ values by inverting these models, as depicted in Figure 1. The models on the left of the diagram are examples

of those used to predict the relative complex permittivity of the soil ($\varepsilon_{soil}$), as a function of frequency, soil moisture by vol-





ume ($m_v$), texture, composition and surface temperature. (Note that $\varepsilon$ is dimensionless and remains commonly referred to by the deprecated term, dielectric constant (Wave Propagation Standards Committee of the Antennas and Propagation Society, 1997)). The models in the middle of Figure 1 use this to predict a radar backscatter coefficient ($\sigma^0_{soil}$), adding in scattering due to the SAR imaging geometry, polarisation and frequency, and surface topography (quantified by the correlation length and root-mean-square height difference, $H_{rms}$). Models on the right of the diagram are among those used to predict the effects of a vegetation layer to estimate the radar backscatter coefficient that the SAR actually measures. The models in the middle of

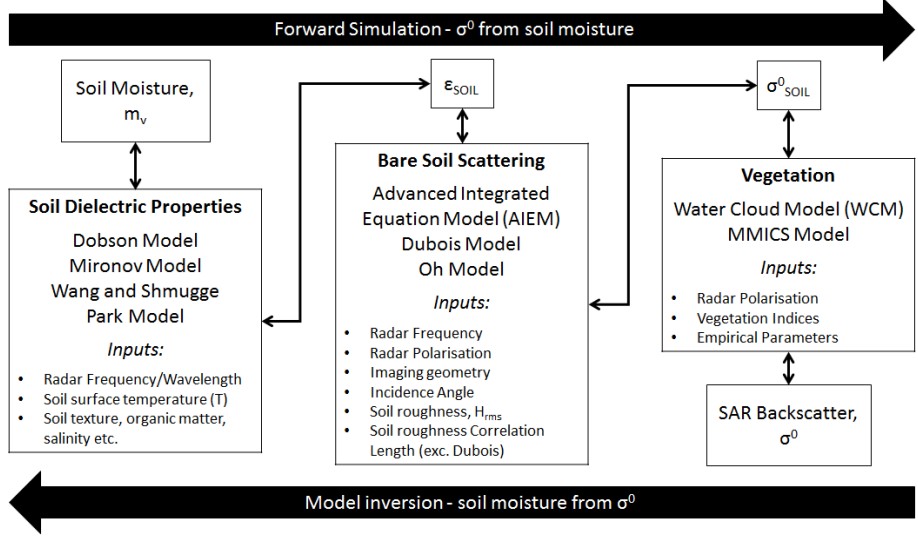

**Figure 1.** Process of estimation of soil moisture (SM) by volume ($m_v$) from SAR backscatter coefficient ($\sigma^0$) through inversion of models

Figure 1, and their alternatives, can be classified into three types:

– Analytical models like the AIEM, are based on theory, may be complex and hard to invert;

– Semi-empirical models (for example the Oh or Dubois models) that have a form which respects some electromagnetic properties, but are simplified with empirically derived parameters;

– Purely empirical models, such as those based on regression.

The calibration of empirical model parameters is typically achieved by fitting a sample of satellite observations and ground measurements (for example, by minimising a cost function as described by Bai et al. (2017)). When analytical model inversion is not possible, it is common to use look up tables or machine learning approaches. Look-up tables are generated from forward modelling using a range of expected input values, one of which will be soil moisture. The inversion is performed by selecting the closest set of input values that would generate the observed output. This method becomes impractical if the range of possible inputs are not constrained. The alternative is to use a machine learning approach to invert the model using forward modelled data as a training data set (El Hajj et al., 2016). A good example of this was recently published by Hajj et al. (2017). Whilst the





training data set in case was very extensive, it was nevertheless generated using calibration parameters for the WCM that were based on winter crops and grassland from two study sites in France and Tunisia (Baghdadi et al., 2017).

As an alternative to model inversion, the same machine learning techniques, such as Artificial Neural Networks (ANN), may also be used in a data driven approach (Paloscia et al., 2008; Notarnicola et al., 2008) . In this situation, models are trained

to provide an optimised transformation between inputs - sensor and ancillary data - and the desired output of soil moisture. Support Vector Regression (SVR) (Pasolli et al., 2011a) is another machine learning technique which can be applied to find a function to estimate soil moisture from SAR data with a certain level of precision of the training data. The models are trained to reproduce ground measurements or simulated data which comprise the training data set. A further independent validation data set is then required to test the performance of the model. The validated model is then applied to new observations to

estimate soil moisture. The benefits of the data driven machine learning approach is that it obviates a detailed understanding of the complex physics and interactions between variables (other than to hypothesise data requirements), and several source data combinations may be quickly evaluated. However, the validity of the model depends on the scope of the training data set. ANNs also have utility in the assimilation of different sensor inputs, for example Sentinel-1 and Landsat8 (thermal infrared (TIR) and NDVI) data (Alexakis et al., 2017), Sentinel-1, SMAP (brightness temperature) and AMSR2 (plant water and soil

surface temperature) (Santi et al., 2018). SVR is particularly suited to multidimensional problems and has been proposed by several authors (Pasolli et al., 2011b; Klinke et al., 2018).

Temporal processing algorithms have been developed to overcome the difficulty in choosing realistic soil surface roughness and vegetation parameters. Time-series analyses (Kim et al., 2012; Pierdicca et al., 2014) have been shown, with bare soil, to reduce the random components of the SAR noise (speckle and clutter) and to provide an estimate of the soil surface roughness.

Change detection algorithms (Wagner et al., 1999; Hornacek et al., 2012; Pathe et al., 2009) assume that the contributions to radar backscatter due to soil moisture, surface roughness and vegetation have different temporal characteristics. In particular, on bare to moderately vegetated, non-arable land, short timescale changes in backscatter coefficient are assumed to be due to changes in SM. A long time-series of SAR data is initially normalised to a local incidence angle. Assuming the range of radar backscatter values represents a complete range of soil moisture conditions between the permanent wilting point and field

capacity, a sensitivity raster is calculated. This is used to calibrate new SAR backscatter images to provide an estimate of the relative soil moisture value at each pixel. Data in this form may be directly useful for many agricultural requirements, but the absolute value of volumetric soil moisture can only be calculated if the field capacity and permanent wilting point of the soil at each pixel is known. This may be obtained, (for example in the UK) from LANDIS soil texture maps and Horizon Hydraulics data (Hollis et al., 2015). The attraction of these approaches is that they do not require prior knowledge of the vegetation

parameters or the soil roughness, and the measurement of relative soil moisture might be more relevant to some potential users of the data, as they are often expressed in relative units ranging from 0% (dry) to 100% (wet) describing the degree of saturation of the surface soil layer (Wagner et al., 1999). However, the change detection algorithm assumes that factors such as surface temperature are negligible, and that vegetation and surface roughness effects are relatively stable over time. Both should be questioned in the context of arable fields. The growth and ripening of crops is quite rapid – over a scale of weeks to months,

and harvesting will cause a step change in vegetation cover over a very short time. For example, wheat and barley crops can





show a change from NDVI < 0.3 to NDVI > 0.8 in around 100 days during growth, declining rapidly before and during harvest (Nasrallah et al., 2018). Step changes in soil roughness may be induced by tillage operations such as ploughing, harrowing, drilling and rolling. The surface roughness of bare soil may also change relatively quickly over time due to erosion by weather, wind and raindrop impact. These assumptions are unlikely to be valid in the context of agricultural fields under crops and

subject to tillage.

Validation of soil moisture estimation processes is usually achieved by regression analysis of estimated soil moisture against ground measurements, using as figures of merit coefficients of determination ($R^2$) and root mean square error (RMSE). The residual errors in these results will be due in part to sensor noise (speckle) but also variability due to the factors that have been approximated (such as surface roughness).

## 3  The Importance of Soil Properties

In this section, we consider the influence of the main properties of soil on the dielectric properties (relative permittivity) of wet soil, and the main factors that were listed in the previous section as making a contribution to the radar backscatter coefficient.

### 3.1  Texture

The soil texture is classified according to its composition in terms of the volume fractions of clay, sand and silt. The effect on

relative permittivity, soil surface roughness and surface temperature effects will now be discussed. While soil properties are likely to have some impact on the vegetation biomass, there is very little evidence to suggest there is an exploitable relationship (Hironaka et al., 1990). However, it has been discovered (Farrar et al., 1994; Demattê et al., 2017) that the increase of NDVI over time following rainfall is dependent on soil texture and chemical composition, with clay soils and aluminium-rich soils showing the greatest rate of increase in NDVI. It is currently unknown how useful these facts would be in the retrieval of soil

moisture from SAR.

### 3.1.1  Relative Permittivity

Following correction for the effects of vegetation, surface roughness and local incidence angle, the relative permittivity of the soil, $\varepsilon_{soil}$, may be determined. From this it is possible to estimate the soil moisture ($m_v$) by inverting a soil dielectric mixing model. Such a model would normally be used to predict $\varepsilon_{soil}$ by combining the relative permittivity values of the constituent

parts of soil, including the solid matter ($\varepsilon_{sand,clay,silt}$), air ($\varepsilon_{air}$) and the water in the soil ($\varepsilon_{water}$). The latter is dependent on how the water is held – whether it is "free", within the large soil pores ($\varepsilon_{free}$) or "bound" to the soil particles by Van der Waals or capillary forces ($\varepsilon_{bound}$) (Park et al., 2017; Wang and Schmugge, 1980) . The models commonly used are those proposed by Hallikainen et al. (1985), Dobson et al. (1985) and Mironov et al. (2004). The latter two were compared by Mialon et al. (2015) and found to produce differences in relative permittivity that are soil texture dependent. More recently a multi-phase dielectric

mixing model by Park et al. (2017) (hereafter referred to as the Park model) has been proposed and validated which accounts





for soil moisture above the saturation point as well as below the permanent wilting point. According to the Park model, $\varepsilon_{soil}$ is a function of the following parameters:

- – Soil moisture by volume, $m_v$

- – Soil texture (proportions of sand, silt, clay), $v_{sand}, v_{silt}, v_{clay}$

– Soil permanent wilting point $m_{vwp}$ and porosity, $p$ (saturation point)

- – Soil temperature, $T$

- – Soil water salinity, $S$

$v_{sand}, v_{silt}, v_{clay}$ and $p$ may be obtained from soil texture maps and $m_{vwp}$ is obtained or derived from soil hydrological data and pedotransfer functions. In earlier work (Wang and Schmugge, 1980) the relative permittivity of a range of different soils

were modelled and verified against laboratory measurements (Wang et al., 1978) ; these results are shown in Figure 2. The results for four of the soils (1-4) are consistent with the Park model, for the others it is likely that the soils were not wetted beyond their saturation point. The soil types referred to are listed in Table 1 (adapted from Wang et al. (1978); Wang and Schmugge (1980)) with the texture class being determined by the textural triangle (Koorevaar et al., 1983). From Figure 2

**Table 1.** Soil types key for Figure 2. The soils are from Arizona and Texas, used for measurements of dielectric constants at 5GHz (Wang et al., 1978; Wang and Schmugge, 1980)

| Soil | Type | $v_{sand}$ % | $v_{silt}$ % | $v_{clay}$ % | Texture Class |
|------|------|------|------|------|------|
| 1 | M5 | 88 | 7.3 | 4.7 | Sand |
| 2 | F2 | 56 | 26.7 | 17.3 | Sandy Loam |
| 3 | H7 | 19.3 | 46 | 34.7 | Silty Clay Loam |
| 4 | HARLINGEN | 2 | 37 | 61 | Clay |
| 5 | YUMA | 100 | 0 | 0 | Sand |
| 6 | VERNON | 16 | 56 | 28 | Silty Clay Loam |
| 7 | MILLER | 3 | 35 | 62 | Clay |

it would be possible to estimate the range of possible soil moisture values $m_v$ that could be represented by a particular value

of $\varepsilon_{soil}$ if there is no information available about the type of soil present. However, the range of soil textures represented by this experimental data is not very broad. To achieve a more comprehensive range of data, it was necessary to use a validated model to determine a relationship between the real part of the relative permittivity and soil moisture, using, as an example, six soil types found in the UK as defined by Table 2. These soil types were chosen to represent the range of variability within the textural triangle and included one (Grove) that sits in the centre of the triangle and could represent a default assumption.

The model proposed by Park et al. (2017) was selected as a recently developed example of a dielectric mixing model that was successfully validated against experimental results. The model considers the contributions of the dielectric properties of the





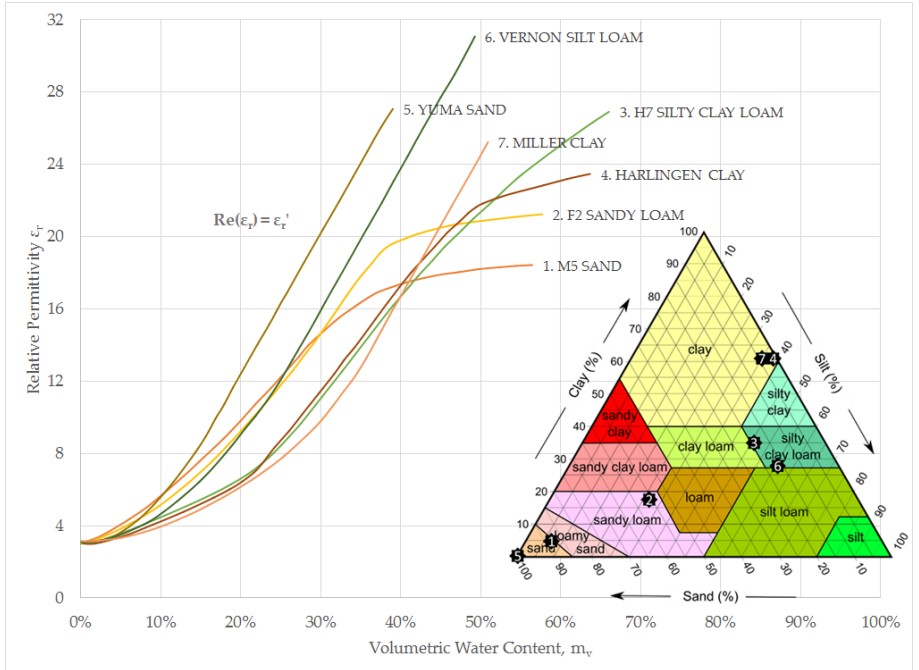

**Figure 2.** Relative permittivity (real part) of several USA soils as a function of soil moisture at 5GHz (C-band) - adapted from (Wang and Schmugge, 1980). The soils are located on the USDA soil textural triangle, inset (graphic credit Mike Norton)

constituents of soil, including air, solid particles (silt, sand, clay), and water. Water can either be free, or bound to clay particles (this modifies the dielectric properties). The model considers three regimes; the first (equation 1) is when the soil moisture is below the wilting point, where all of the water is assumed to be in the bound state, the second (equation 2) is when the soil moisture is above the wilting point but below saturation level, where there is a mixture of bound and free water, and thirdly

5 when the soil is above saturation point, when there is excess free water that cannot be held by the soil and is generally held on the surface but cannot immediately drain away. This, third (equation 3) situation was a new consideration when compared to previous models, but one that is important for flood prediction. The following equations from the Park model were used to calculate the real part of the relative permittivity of the soil ($\varepsilon'_{soil}$) as a function of soil moisture ($w$), porosity ($p$), wilting point ($w_{wp}$), the volume fractions of sand, silt and clay ($v_{sand}$, $v_{silt}$, $v_{clay}$) and the relative permittivities of the soil constituents

10 ($\varepsilon'_{sand}$, $\varepsilon'_{silt}$, $\varepsilon'_{clay}$, $\varepsilon'_{bound}$ (water), $\varepsilon'_{free}$ (water) and $\varepsilon'_{air}$):

For $w \leq w_{wp}$ :

$$\varepsilon'_{soil} = 0.8\big((1-p)(v_{sand}\varepsilon'_{sand} + v_{silt}\varepsilon'_{silt} + v_{clay}\varepsilon'_{clay}) + m_v\varepsilon'_{bound} + (p-m_v)\varepsilon'_{air}\big) \tag{1}$$

For $w_{wp} < w \leq p$ :

$$\varepsilon'_{soil} = 0.8\big((1-p)(v_{sand}\varepsilon'_{sand} + v_{silt}\varepsilon'_{silt} + v_{clay}\varepsilon'_{clay}) + m_v\left(\left(\frac{p-m_v}{p-m_{vwp}}\right)\varepsilon'_{bound} + \left(\frac{m_v-m_{vwp}}{p-m_{vwp}}\right)\varepsilon'_{free}\right) + (p-m_v)\varepsilon'_{air}$$





$$(2)$$

For $w > p$ :

$$\varepsilon'_{soil} = 0.8\big((1 - m_v)(v_{sand}\varepsilon'_{sand} + v_{silt}\varepsilon'_{silt} + v_{clay}\varepsilon'_{clay}) + m_v\varepsilon'_{free}\big) \tag{3}$$

The values for $\varepsilon'_{free}$ and $\varepsilon'_{free}$ were determined as a function of radar frequency ($f$), temperature ($T$) and salinity ($S$) using

the Debye relaxation formula by the method set out by equations 25-31 in (Park et al., 2017), setting both $\varepsilon^{min}_{free}$ and $\varepsilon^{min}_{bound}$ to the value 4.9. The soil parameters were obtained from the LANDIS Horizon Hydraulics and Horizon Fundamentals data sets (Hallett et al., 2017). In each soil series, the A soil horizon typically found at the surface, under arable land use, was chosen. The frequency, $f$ was set to 5.4 GHz, the salinity, $S$, was set to zero, and the soil temperature, $T$ set to 10°C (this is a typical value in the UK).

**Table 2.** Input parameters used for simulation in the Park et al. (2017) model. Data is from six UK soil series, representing typical soil characteristics in the upper soil horizon under arable land use. Source of data is LANDIS (Proctor et al., 1998; Cranfield University, 2017).

| Soil Series | Type | $w_{wp}$ % | $p$ % | $v_{sand}$ % | $v_{silt}$ % | $v_{clay}$ % |
|---|---|---|---|---|---|---|
| Wensum | Clay | 35.7 | 66.6 | 3 | 26 | 71 |
| Sollom | Sand | 8.5 | 57.7 | 94 | 5 | 1 |
| Munslow | Silt Loam | 18.4 | 51 | 4 | 80 | 16 |
| Grove | Clay Loam | 24.4 | 54.1 | 32 | 34 | 34 |
| Stixwould | Clay | 31.4 | 62.2 | 18 | 27 | 55 |
| Wyre | Clay | 29.9 | 61.1 | 20 | 32 | 48 |

The output of the model (Figure 3) shows that there are considerable variations between the different soil textures. High clay-content soils are predicted to have a lower relative permittivity, due to a significant proportion of the soil water molecules being bound to clay particles, with electrical properties modified with respect to those of free water that dominates in sandy soils.

Figure 3 illustrates the variability in soil moisture with soil texture, based on a given observation of $\varepsilon'$. The curve for each

soil texture was generated by reversing the Park model - iterating through values of $m_v$ to find the one that yields a particular value of $\varepsilon'$. The range of potential soil moisture values for a particular relative permittivity is shown by the dashed line, which lies mostly between 10% and 20% water by volume. This represents the limits of spatial variation in soil moisture, for a given backscatter value, due to soil texture alone.

### 3.1.2  Soil Surface Roughness Scattering

Surface roughness is a major factor in soil moisture estimation from SAR, especially in arable fields subject to anthropogenic manipulation. The step changes in soil roughness induced by tillage operations such as ploughing, harrowing, drilling and rolling, are illustrated by Figure 4, which is based on profiles published by Evans (1980) and Martinez-Agirre et al. (2016).





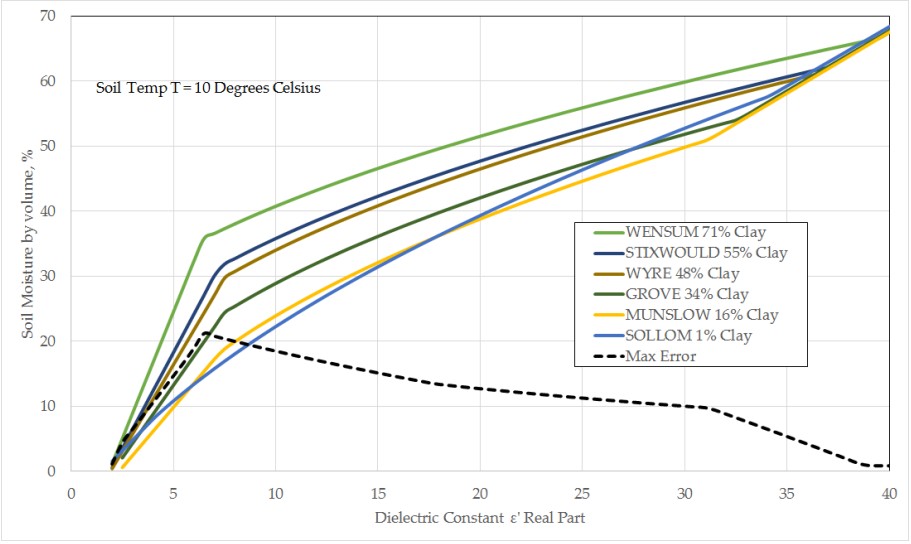

**Figure 3.** Soil moisture, $m_v$, as function of the real part of the relative permittivity, $\varepsilon'$ at 5.4 GHz, for six UK soil types, as predicted by reversing the model of Park et al. (2017)

The description of Morgan (2005) is consistent with this data, which shows that primary tillage (ploughing) results in a rough, sometimes cloddy surface with $H_{rms}$ values of up to 6 cm. Secondary tillage (harrowing) reduces this to 1-3 cm, which is often further reduced by drilling and rolling. There is potential for roughness changes of up to 5 cm in $H_{rms}$ between image acquisitions.

The assumptions behind soil moisture retrieval by change detection are not valid within fields subject to regular tillage or harvesting and should, at least, be flagged. Tillage events could be detected from sudden changes in backscatter within field boundaries that are not typical of their wider spatial context. It may be possible to go further and compensate for the change in $H_{rms}$ that is expected as a result of a tillage operation, based on the soil texture and the field's temporal and agricultural context.

The review paper by Zobeck and Onstad (1987) compared the measurements of soil Random Roughness (RR) (Romkens and Wang, 1986) as a function of tillage operations and soil texture. To obtain enough points for a simple regression analysis (Figure 5) other points have been added from data extracted from soil surface profiles published by Currence and Lovely (1970), Evans (1980) and Martinez-Agirre et al. (2016) (where a clay content of 35% has been estimated from the textural triangle for silty clay-loam). The data points have been categorised by the type of tillage operation as Primary, Secondary or

No Till. The extended data, plotted in Figure 5, provides some evidence of a correlation between soil roughness (RR) after primary tillage and soil clay content, with an $R^2$ value of 0.32. There is no apparent correlation between the soil texture and RR after secondary tillage operations, but there is an apparent correlation for No Till with $R^2 = 0.37$. The cloddy nature of a ploughed clay soil is commonly observed in agricultural fields; contrasting with the same treatment of a sandy soil in Figure 6. In a recent study by Martinez-Agirre et al. (2017) a number of alternative soil roughness parameters were compared for their





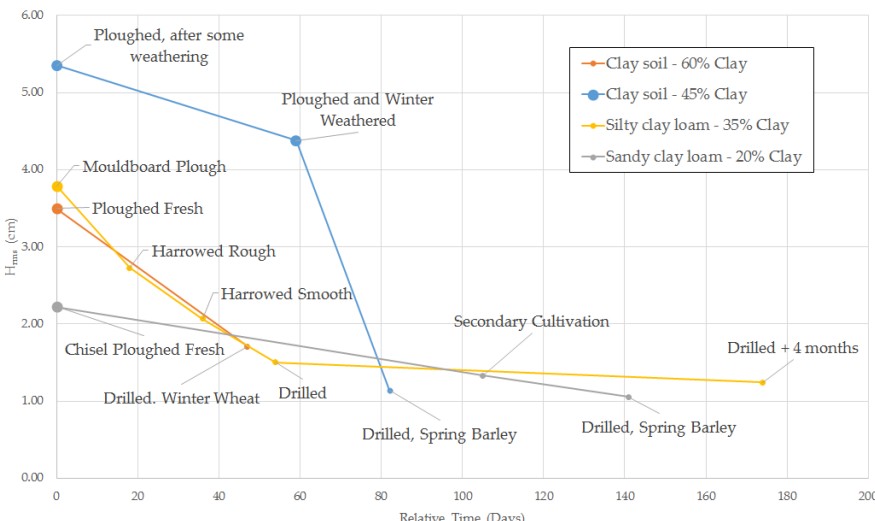

**Figure 4.** Evolution of soil roughness over time and due to tillage for several soil textures. Larger dots are primary tillage operations, small dots secondary tillage. Data taken from Evans (1980) and Morgan (2005).

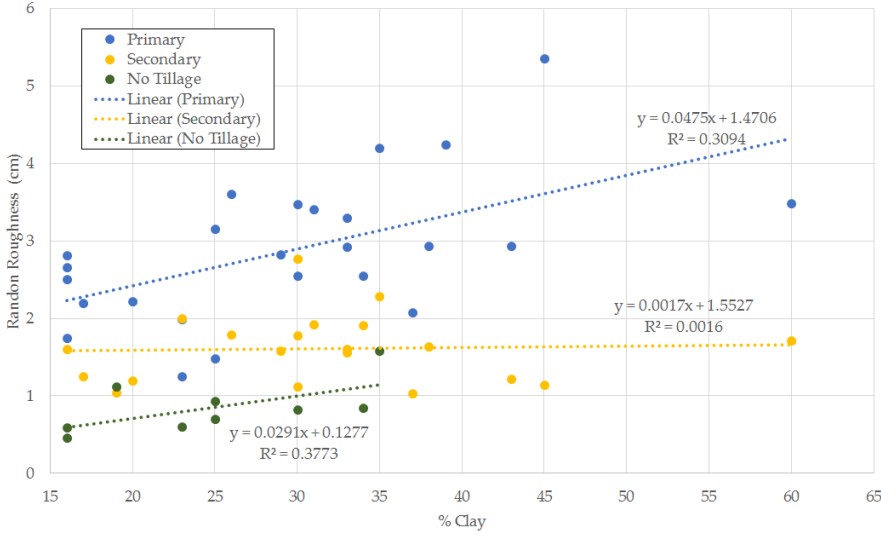

**Figure 5.** Soil surface roughness ($H_{rms}$) as a function of clay content for tillage operations. Data from Currence and Lovely (1970); Evans (1980); Martinez-Agirre et al. (2016); Zobeck and Onstad (1987).

ability to predict the effect on C-band SAR backscatter coefficients for a range of tillage operations. The conclusion was that the peak frequency $F$ (Romkens and Wang, 1986) (the number of elevation peaks per unit length) and fractal dimension $D$ (Shelberg et al., 1983; Vidal Vázquez et al., 2005), were most strongly correlated to $\sigma^0$ and provided the best class separation





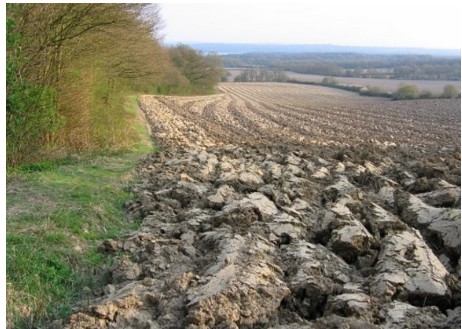
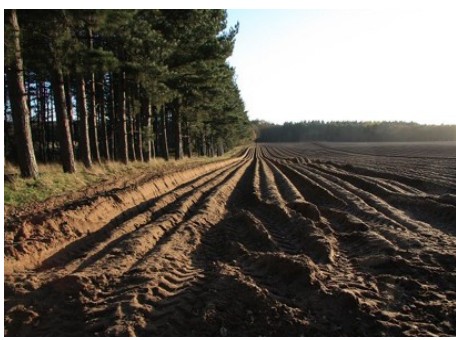

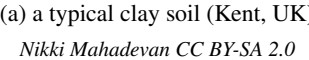

(a) a typical clay soil (Kent, UK)

*Nikki Mahadevan CC BY-SA 2.0*

(b) a sandy soil (Suffolk, UK)

*Bob Jones CC BY-SA 2.0*

**Figure 6.** Examples of freshly ploughed soil roughness

between the various tillage operations. The fractal dimension $D$ measures surface geometric complexity is a number between 2 (smooth surface) and 3 (maximum roughness) and is calculated by several methods (Vermang et al., 2013). This suggests that, if some means can be developed to detect $F$ or $D$, there is potential to classify agricultural fields by tillage state, and then use this information within the estimation algorithm, by using $H_{rms}$ values typical of the soil type and tillage state.

5    Another factor related to soil roughness is depression storage or ponding of standing water when the soil is saturated or immediately after heavy rainfall. The smooth surface of the ponded water will act as a specular reflector for microwaves, so the backscatter coefficient for any area including ponded water will be significantly reduced. Whilst larger areas of standing water may be detected and masked in a SAR image, small pools of water will be sub-pixel, and contribute to an underestimated soil moisture. The depression storage capacity of clay soils is 1.6 to 2.3 times that of sandy soils according to Morgan (2005), 10    and is also linked to surface roughness, slope and furrow orientation (Edwards et al., 1994). This is therefore another soil texture-dependent variable.

In summary of the findings above, knowledge of the expected sequence of farming operations in conjunction with soil texture maps can inform the choice of soil roughness ($H_{rms}$) parameters as follows:

– $H_{rms} \approx 0.5$ to $1.0$ cm for all soil types under zero tillage;

15    – $H_{rms} \approx 1$ to $2$ cm for all soil types after secondary tillage and drilling;

– $H_{rms}$ is weakly correlated with % clay content after ploughing, ranging from 2 to 6 cm;

– $H_{rms}$ reduces to 1 to 2 cm due to erosion and secondary tillage over a few months;

– $\Delta H_{rms} \approx 1$ to $5$ cm on ploughing in the spring or autumn between image collections, depending on the clay content of the soil and the type of plough used.





## 3.2 Soil temperature

There is a need to assess the potential impact of soil temperature on the soil moisture estimation. The Park Model has been employed in our study to assess the dependency of the relative permittivity of the soil with surface temperature for a range of soil moisture values. Using the example of soil type Grove the real part of the relative permittivity was modelled across a range of surface temperatures, then repeated for different values of soil moisture (Figure 7(a)). This shows that, for soil moisture less

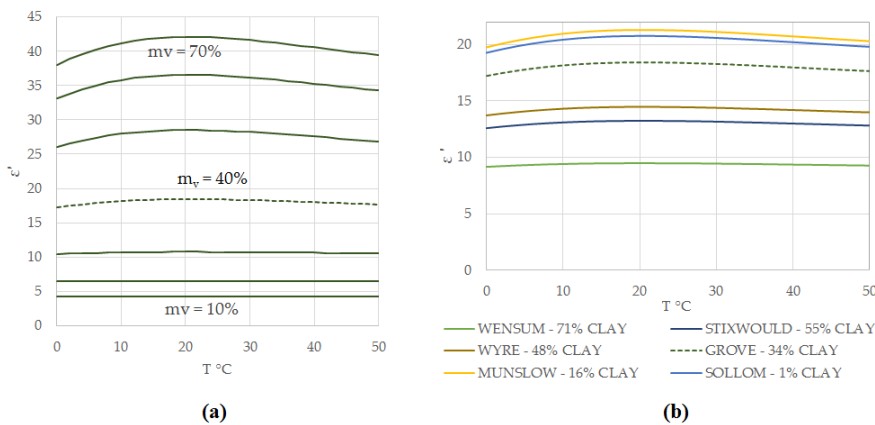

**Figure 7.** Variations in predictions of the real part of the relative permittivity surface temperature, $f = 5.4$ GHz for: (a) a range of soil moisture values and Grove topsoil; (b) a range of soil textures (by clay content) and $m_v = 40\%$. The dashed line is the same data (Grove at $m_v = 40\%$)

than about 30% by volume and this soil texture, there is little or no temperature dependence. However, at higher soil moisture values there is a greater temperature dependence. At these higher soil moisture values, an error of up to 5% by volume could be introduced by not accounting for soil surface temperature in the retrieval process.

Figure 7(b) shows that this temperature dependence is more enhanced in sandy soils, and less important in clay soils. This is confirmed in a recent study by Rodionova (2017a) who found a significant temperature dependence on the radar backscatter coefficient measured by Sentinel-1, but only on one site with a very sandy soil. The high sand content soils that are shown to exhibit significant temperature dependent effects soils would be beyond saturation under the conditions of Figure 7(b), which is a rare condition. For most soils and typical soil moisture ranges the temperature dependence would be very small.

The Park model does not yet include the effect of freezing on the soil relative permittivity, but this also very significant, as documented by several papers (Zhang et al., 2003; Mironov et al., 2017b; Baghdadi et al., 2018a; Park et al., 2011; Mironov et al., 2017a). A typical relationship between soil relative permittivity and temperature is shown in Figure 8. As the temperature falls, there is a sharp decrease in $\varepsilon_r'$ at the freezing point of water, with the relative order of sand, silt, clay being inverted at this point. Knowledge of the soil texture would help confirm that the change is due to freezing and to correct for it. In this example, the decrease in relative permittivity caused by freezing is equivalent to a drop in volumetric moisture content from over 30% to under 15%. This creates an ambiguity in locations where diurnal freezing of the topmost soil layer is common in





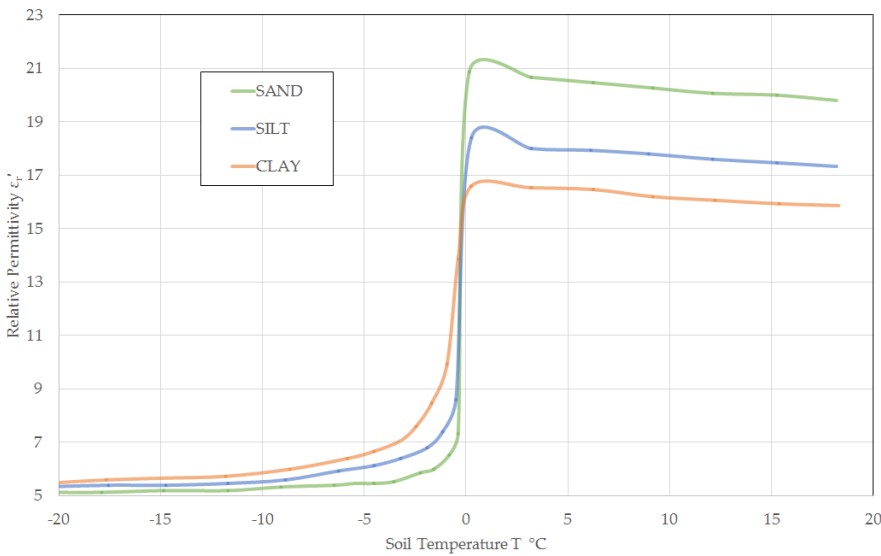

**Figure 8.** Variation of real part of relative permittivity with temperature for sand, silt and clay at $m_v$=30%, adapted from Zhang et al. (2003).

winter. A context sensitive or time-series analysis will be required to detect and correct for this source of error. The potential of using this effect to detect and map frozen soil with Sentinel-1 data was recently demonstrated by Baghdadi et al. (2018a), who observed a reduction of about 2 dB in $\sigma^0$ between unfrozen and frozen soil.

### 3.3 Soil Organic Matter

The presence of organic matter or humus in the soil has received relatively little attention in the context of SAR remote sensing of soil moisture at C-band. Experimental results are sparse and appear contradictory. Some studies show that microwave dielectric constant increases slightly with humus and organic carbon (Jackson, 1987; Chaudhari, 2015), whilst other suggest the opposite trend (Bobrov et al., 2005; Manns et al., 2015). This apparent conflict may be explained by measurements on 12 soil samples carried out by Liu et al. (2013), who also explored a range of soil moisture conditions. Low soil organic content

(SOC) demonstrated a behaviour similar to that predicted (for its texture) by the Park model as shown by the blue curve in Figure 9. A high SOC soil (with the same soil texture, orange curve) was found to have lower relative permittivity when dry. At $m_v \approx 40$% the lines cross such that the high SOC soil has a higher relative permittivity, as shown by Figure 9. This suggests that the high SOC sample is behaving as if it had a texture with a higher fraction of clay, with a greater proportion of bound water and a higher soil moisture at saturation (above 55%). The confusion in some published results may be as a result of taking

measurements with a moisture content at which the lower SOC samples are saturated. It is also possible that, if the humus is very coarse in texture it might not behave like clay, but perhaps as sand.

    The example of Figure 9 suggests variability of up to 10% in $m_v$ could be caused by spatial changes in SOC, which means that it is as important as soil texture.



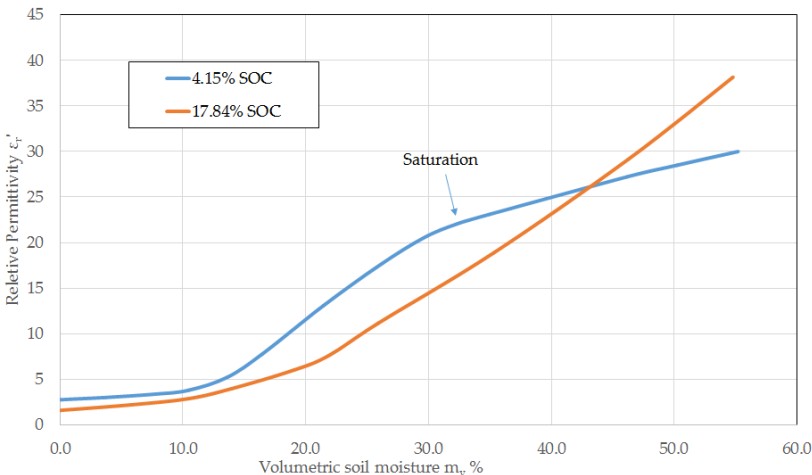

**Figure 9.** Dependence of relative permittivity with soil moisture for two soils with significantly different organic matter content. This is a simplified version of data presented by Liu et al. (2013) of measurements on soil samples from Heilongjiang Province, northeast China.

### 3.3.1 Salinity

Soil salinity is a major problem in certain areas, due to the build-up of minerals in the soil due to evaporation of groundwater, irrigation and reduced leaching due to low precipitation. There is considerable interest in mapping areas of saline soils, and recent research by Taghadosi et al. (2018) has examined the potential of using C-band SAR data to do this. As the relative

permittivity of saline water differs from pure water, the relative permittivity of soils is expected to vary with salinity. Simulations have been undertaken using dielectric mixing models and seawater dielectric models. Recent work by Li et al. (2014) compares such simulations with laboratory measurements has concluded that this method does not accurately predict the effect of salinity on radar backscatter. However, it seems that the real-part of the dielectric constant is less dependent on salinity than the imaginary part, but the effect is more significant at high soil moisture values and that the C-band VV polarization is quite

sensitive to the effect. In earlier work Jackson and O'Neill (1987) concluded that for moderate salinity values, consistent with unaffected plant growth, salinity could be disregarded in the microwave emission of the soil. Nevertheless, salinity remains a potential source of error in soil moisture retrieval, where soils have significant salinity issues, and that error is currently difficult to quantify until further models are developed.

### 3.4 Soil Heterogeneity

In practice, all soils exhibit horizontal and vertical heterogeneity, due to its structure (aggregates and pores), the presence of stones, and local variations in texture, organic matter, drainage, compaction and many other factors. Jackson and O'Neill (1986) investigated the effect of soil structure, finding that the dielectric mixing models did not predict, very accurately, the relative permittivity of soil that contained clods or significant pockets of air (such as after ploughing) even if the roughness was





correctly characterised. This was said to be because the models assume, unrealistically, a homogeneous three-phase mixture of minerals, air, and water. The effects of rock fragments has also been investigated by Jackson et al. (1992) who found them to be very significant at 5GHz, where at 35% rock content the dynamic range of radar backscatter with soil moisture was very low. There is, however, a link between the rock and stone proportions in soil and soil roughness that could be exploited for

prediction. Unfortunately the heterogeneity of soil type and other properties may not be described at an appropriate resolution by soil maps, as examined by Lee et al. (2017). Stones and air pockets are therefore another cause of errors in SM estimation.

Vertical heterogeneity was studied by Gorrab et al. (2014) who found it to be a significant issue at C and X band (resulting in additional noise of 1 to 1.7 dB), where the penetration depth varies according to soil moisture between approximately 1cm (wet, $m_v$ = 30%) and 3cm (dry, $m_v$ = 7%) (Nolan and Fatland, 2003). Soil properties may vary significantly in the top 3 cm.

Further vertical heterogeneity in soil moisture can also occur following rain, due to time lags due to infiltration (Gorrab et al., 2014). Failure to account for these effects is a further source of error in soil moisture estimation.

## 4 Discussion

### 4.1 Summary

We have shown that, for a given SAR backscatter value, there is variability in soil moisture estimation that is directly or

indirectly driven by soil properties (texture, structure, organic matter content, salinity and stone content). These are summarised in Table 3. Some or all of these soil properties may be obtained from existing soil maps and databases, chosen to match the

**Table 3.** Impact of soil properties on factors that affect soil moisture estimation from SAR: 0 = no impact to 4 = high impact; $W_{wp}$ – Wilting point, $W_{fc}$ – Field Capacity.

| Soil Property | $W_{wp}, W_{fc}$ | $H_{rms}$ | $\varepsilon(m_v)$ | $\varepsilon(T)$ | $\Delta\varepsilon$(On freeze) | Dew |
|---|---|---|---|---|---|---|
| Texture | 4 | 3 | 2 | 1 | 2 | 2 |
| Organic Matter Content | 3 | 1 | 2 | 1 | 2 | 2 |
| Salinity | 0 | 0 | 1 | 1 | 1 | 0 |
| Stone content | 2 | 4 | 2 | 0 | 0 | 1 |
| Structure | 2 | 3 | 2 | 0 | 0 | 0 |

field scale where possible. Whilst the Harmonized World Soil Database (Fischer et al., 2012) has a resolution of 1km, national and regional maps are often available at higher resolution, especially in agricultural areas. As a basis on which to estimate vegetation cover or soil roughness, land cover maps may be useful. As an example, the Corine land cover map of Europe

(European Environment Agency, 2006) has a minimum feature size of 25 hectares, and limited cover classes, but higher resolution local maps are often available. For example the UK Centre for Ecology and Hydrology (CEH) has published its *Land Cover plus: Crops* product (Jarvis et al., 2019), which categorises two million land parcels into annual crop types for every field in Great Britain. Potential agricultural stakeholders in soil moisture estimation by remote sensing are likely to





have accurate soil type and land use information for their areas. There is a mostly, as yet, unexplored potential to exploit this information to improve the confidence and relevance in the estimation process. For example, in the case of soil roughness, where there is no *in situ* or remotely-sensed measurement, it may be possible to predict it over time using a model, which might take, as inputs, the crop being grown, its stage of development, the soil properties, rainfall and a time-line of tillage

operations in the field. It is also possible to use the SAR data itself to detect tillage and harvesting operations, techniques such as this are already used to compile products such as the (CEH) *Land Cover plus: Crop* maps. Harvest date detection would be valuable for the purpose of making corrections for crop cover.

Soil temperature may be more important than has been recognised so far, particularly with respect to dew formation and temporary freezing of the soil. Where they occur, surface frosts and dew are routinely forecasted by meteorological services.

A method for identifying and correcting for areas by dew and frost is desirable.

## 4.2 Implications for SM Estimation Algorithms

With the current level of interest in using machine learning approaches to develop models to estimate soil moisture from SAR data, it is important to emphasise that errors are likely to occur if such models are trained against unrepresentative point ground data in one locality against a limited set of land uses and soil properties, then these models are used in a more general context.

One particular issue that is often overlooked is that the SAR only responds to moisture in the top few centimeters of soil, whereas the ground data is often collected at greater depth (tens of centimeters). To apply such models more widely they need to be trained on input data at an appropriate depth and that describes all of the sources of variability that have been identified in this study.

It is evident that simple change detection algorithms cannot be expected to perform well at field scale in areas of arable

cultivation. The underlying assumption that short term variation in radar backscatter is dominated by soil moisture does not hold if soil surface roughness and vegetation biomass are being managed anthropogenically. Rapid changes in backscatter may also occur diurnally in many areas due to the formation of dew, frost and soil surface freezing. Some of these step changes may be detectable in the remotely sensed data or may be otherwise predicted and assimilated into a more complex algorithm to apply appropriate corrections, or at least flag greater uncertainty. Soil properties obtained from soil maps may be used to

obtain absolute soil moisture estimates from the relative values normally produced by change detection algorithms, however this calculation should take account of organic matter and stone content, where this information is available.

## 5 Conclusions

This study set out to identify the sources of uncertainty in soil moisture estimation from SAR, in the context of C-band sensors on satellites such as ESA's Sentinel-1, and assess whether knowledge of soil properties could, potentially, be used to reduce

these errors.

Soil roughness, $H_{rms}$, in agricultural fields subject to tillage is subject to large changes during the year, and cannot be assumed to be slowly varying compared to soil moisture changes. It may be estimated, in principle, by combining detailed





land use data, the temporal context of the measurement within the expected sequence of farming operations, and soil texture information. The soil texture appears to be an important factor in a model for estimating roughness after primary tillage, such as ploughing.

The effect of vegetation, though very significant, has not been the main focus of this study. In the case of agricultural fields
large temporal changes in vegetation biomass can also be expected during the year. It may be possible to use SAR data to detect major changes in vegetation cover and infer that a farming operation has taken place. This information may be useful in correcting for the vegetation effects on such fields.

Leaving aside these two major factors, the study quantified the variability in SM estimation due to soil texture (up to 20% soil moisture by volume) and organic matter (a further 20%). Other sources of variability, not yet quantified, include salinity,
structure and heterogeneity.

Soil surface temperature normally has a minor impact on soil moisture estimation in non-frozen soils contributing to a variability of only 5% by volume in the SM value, and only then if the soil is very wet and sandy. Frozen soil has very different radar backscatter characteristics; without correction for this, a much lower value of SM will be estimated. The error (up to 40% in $m_v$) is soil texture dependent. Dew is likely to be a significant problem in some locations, and it could have a significant
impact on results.

Many of these issues are likely to be factors that are significant in many agricultural areas. Algorithms that have been developed and validated elsewhere may require further refinement.

## 6  Recommendations

This study has shown that soil texture and other properties can have a significant impact on SM estimation from C-band
SAR. It would be logical to next assess the improvement in the performance of model inversion and data driven machine learning approaches when trained with additional data based on reliable and high resolution soil property maps, and with ground measurements taken at an appropriate depth in the soil. Further research should also focus on several areas that drive uncertainty in SM estimation from C-band SAR in agricultural areas. Soil surface roughness, crop and vegetation effects and temperature effects (dew and freezing soil) are perhaps the most significant, but the effects of organic matter content, soil
heterogeneity and salinity are worthy of more attention, too.

Soil surface roughness correction might be improved by combining remote sensing estimation of roughness parameters with prediction and modelling. To develop a model to predict soil roughness in agricultural areas, the relationships between soil surface roughness and agricultural processes, crop type and stage of development, soil properties and topography should be explored further. The remote sensing element of soil roughness estimation is also a priority, in particular the use SAR data
detect step changes in radar backscatter due to tillage and harvesting. Correction for dew and freezing soil is also important in many areas, prediction from meteorological forecasting may be of some assistance but is unlikely to have the spatial or temporal resolution. Ideally, a remote sensing solution, preferably using the SAR data itself, will be needed to identifying and map areas of dew or frozen soil.



There is potential to extend change detection algorithms to perform as well in agricultural fields as elsewhere. This would be possible once reliable methods are established to detect and quantify sudden changes in radar backscatter due to freezing soil, dew, and the impact of farming practices on soil roughness and vegetation cover.

Lastly, given that the radar backscatter is dependent on soil properties, there is exciting potential, yet to be fully explored, in using SAR data to improve the spatial resolution and mapping accuracy of soil properties, including soil texture, organic matter content and salinity.

*Author contributions.* John Beale conducted all of the research and analysis and wrote the paper. Dr. Toby Waine, Prof. Ron Corstanje, Dr. Boris Snapir and Dr. Jonathan Evans provided guidance and a technical review.

*Competing interests.* The authors declare no conflict of interest. The funding sponsors had no role in the design of the study; in the collection, analyses, or interpretation of data; in the writing of the manuscript, or in the decision to publish the results.

*Acknowledgements.* This work was supported by the Natural Environment Research Council [grant number NE/M009106/1], and the Biotechnology and Biological Sciences Research Council. The authors would like to acknowledge the support of the Soils Training and Research Studentships Centre for Doctoral Training (STARS CDT). Thanks are also due to Chang-Hwang Park for assistance in the implementation of his model, and to Prof. Dick Godwin for identifying earlier research into tillage and depression storage.



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
