# Peer review of "The significance of soil properties to the estimation of soil moisture from C-band synthetic aperture radar"

_Hydrology and Earth System Sciences, 2019_

## Referee Comment (RC1) · Anonymous Referee #1 · 19 Jul 2019

The authors discuss based on literature the effect of soil properties on the C-band SAR observations and extrapolate this to the estimation of the soil moisture. They have chosen C-band because Sentinel-1, as the first operational space borne SAR system, provides C-band measurements. Yet, the authors do not analyse a single Sentinel-1 SAR image, while they are downloadable.

Although the manuscript is reasonably well written, the description of many elementary aspects of SAR remote sensing of soil moisture is incomplete, some examples are given under the detailed comments. Further, I do not really understand the prime focus on soil properties as a source of uncertainty affecting SAR-based soil moisture

retrieval. The issues of radiometric accuracy of Sentinel-1, surface roughness parameter uncertainty and uncertain vegetation effects are much more pressing at field scale. The manuscript in it is current form is a review paper that does not adequately describe the state of the art in the field and is without any original research of little added value to the community.

Detailed comments:

General: The abbreviation for soil moisture is not consistently used. Sometimes soil moisture is written out. Sometimes SM is used and on p3l13 mv is used.

P2L17-33: The authors give an ambiguous description of the relationship between soil moisture and microwave (active/passive) observations, emission and backscattering . Soil moisture determines the dielectric constant and the dielectric constant is part of the refractive index that defines the specular reflection of microwave radiation, which is related to both the amount of microwave emission and backscattering.

P2L34: I agree with the authors that there are 'many unknown factors' that influence radar observations. In following sentence, however, the focus is on the soil properties, while from research it is well known vegetation, soil surface roughness and topography are much more important factors. In fact, much research (also cited later on in that manuscript) on how soil properties affect the relationship between soil moisture and the dielectric constant has been done.

P3L31: what do the authors mean by 'take account of incidence angle'? Do they mean that the backscatter is also determined by incidence angle? or that the derivation of the backscatter requires correction for the local incidence angle? Both are actually the case.

P3L31: 'validate' from this sentence it is not clear that it concerns here soil moisture validation. The jump from SAR processing to validation of soil moisture retrievals is for me too big. soil moisture validation.

P4L7: R2

P4L12: 'Vegetation . . .' this needs to be supported by a reference. Actually I think that surface roughness the most important factor and not vegetation.

P4L13: 'attenuating both the transmitted radar signal and the backscatter from the underlying soil'. Can the authors explain what the difference is between the 'transmitted radar signal' and 'the backscatter from the underlying soil'

P4L17: This statement is not correct, and also depends on the incidence angle and vegetation type, see for instance Joseph et al. RSE 2010 or Mattia et al. TGRS 2003. Often the vegetation – soil scattering terms are responsible for a significant amount of soil moisture sensitivity beyond a LAI 0.5 m2 m-2.

P4-5: Apart from the WaterCloud and MIMICS models, there are other models that are nowadays used more frequently by the community, e.g. the distorted born approximation by Lang and Sidhu (1983) was proposed by Kim et al. (TGRS 2013) for the SMAP radar product and the output of the Tor Vergata discrete scatterer model is used for SMOS soil moisture retrieval over forests.

P7L1: It would be better if the authors would state that the dielectric constant is defined as electric permittivity relative to the electric permittivity of free space.

P7: Figure 1 suggests that there different models to simulate the bare soil backscatter, but that there is only the water cloud model to account for the effects of vegetation. This is a bit out of balance.

---

## Referee Comment (RC2) · Anonymous Referee #2 · 9 Oct 2019

General comments:

This is an interesting work investigating potential perturbing factors and "forgotten processes" of soil moisture retrieval from SAR data at C-band. This is useful because some of these limitations are not often presented in the existing literature, and because many applications derived from Copernicus Sentinel-1 data are emerging. From this point of view, the focus on the impact of soil temperature on permittivity is particularly interesting. The overview of recent results regarding soil roughness is also interesting for HESS readers. Comments on the limitation of the change detection approach are relevant. However, this work cannot be considered as a research paper. No new orig-

inal result is presented. Category could be changed to "review paper". To reach this status, a more complete overview of the literature is needed (e.g. recent research on the use of interferometry to retrieve surface properties).

Recommendation: major revisions.

Particular comments:

- P. 3, L. 18 ("Sigma0"): Yes, Sigma0 is useful. What about other quantities such as phase, coherence? The possible use of interferometric information to constrain soil moisture was investigated by a number of authors. E.g. Scott et al. (https://doi.org/10.1038/s41598-017-05123-4) and other works cited by Scott et al.. This cannot be ignored.

- P. 4, L. 16 (LAI): Is LAI the main factor or should vegetation water content or plant biomass be considered?

- P. 6, L. 9 (dew): What about intercepted rainwater?

- P. 9, L. 17 ("increase of NDVI"): This could be a matter of soil water holding capacity. E.g. Dewaele et al. (https://doi.org/10.5194/hess-21-4861-2017) found a relationship between maximum annual LAI values and SWHC for straw cereals.

- P. 10, L. 11 ("for the others..."): Why ? mv values for soils 1-4 (less than 50%) sound more realistic to me than for soils 5-7 (above 50%). mv larger than 50% are usually observed for organic soils presenting a very large fraction of organic matter. Are soils 5-7 organic?

- P. 13, L. 1 (Fig. 3): Are they organic soils? mv values larger than 50% are quite uncommon for mineral soils. Why 10°C? Is a similar result obtained at 25°C for example?

Editorial comments:

- P. 4, L. 7: "2<0.22" ?

---

## Author Comment (AC1) · 5 Nov 2019

The authors thank the referee for the time taken to review this paper, and for the detailed and constructive comments and suggestions.

The authors' response is contained within the supplementary PDF file, which presents the cumulative responses to all referees.

Please also note the supplement to this comment:
https://www.hydrol-earth-syst-sci-discuss.net/hess-2019-294/hess-2019-294-AC1-supplement.pdf

**Supplement:**

**"The significance of soil properties to the estimation of soil moisture from C-band synthetic aperture radar" by John Beale et al.**

We have formatted this document to show, first the referee comment in *italics,* followed by the authors' response in plain text[1]. Where we include propose changes to the manuscript the changes to the wording are underlined.

The authors thank both referees for the time taken to review this paper, and for the detailed and constructive comments and suggestions. The revised manuscript will be improved by addressing these points, as we describe below. Our intention is to describe the impact of soil properties on SAR estimation of soil moisture, and we base this on original research (as detailed under response A2(c) below).

**Reply to Referee #1 ([https://doi.org/10.5194/hess-2019-294-RC1, 2019](https://doi.org/10.5194/hess-2019-294-RC1, 2019))**

General Comments

*The authors discuss based on literature the effect of soil properties on the C-band SAR observations and extrapolate this to the estimation of the soil moisture. They have chosen C-band because Sentinel-1, as the first operational space borne SAR system, provides C-band measurements. Yet, the authors do not analyse a single Sentinel-1 SAR image, while they are downloadable.*

**Response A1**: The paper seeks to explore the physics of the interaction between SAR sensors and the properties of the soil from a theoretical perspective, by analysing, in a new way, data that was collected in previous studies and by exploiting recent models. We agree that an example of a SAR image alongside optical data may be useful for readers less familiar with SAR images of agricultural areas.  In the revised manuscript we will, add a new sentence to the paragraph ending P2L33, followed by a new Figure 1, as indicated below.

"…This study focuses on C-band because data from the European Space Agency's Sentinel-1 satellites is freely available, enabling regular derivation of soil moisture products at the required scale and resolution. Figure 1 shows an example SAR backscatter image from Sentinel-1 of agricultural fields contrasted with a natural colour image from the Sentinel-2 satellite."

[Figure]

*Figure 1 - Example of Sentinel-1 SAR image, VV polarisation (left) of agricultural fields at Morley in East Anglia, UK contrasted with natural colour imagery from Sentinel-2 (right). Date 11th October 2018. At this scale soil moisture would be assumed to be similar for both bare and vegetated fields and yet the backscatter contributions are highly contrasting, demonstrating the potential impact of vegetation, surface roughness after tilling, and soil properties.*
* * *
[1] Numbered sequentially, prefixed by "A" for responses to Referee #1 and "B" for responses to Referee #2.

*Although the manuscript is reasonably well written, the description of many elementary aspects of SAR remote sensing of soil moisture is incomplete, some examples are given under the detailed comments. Further, I do not really understand the prime focus on soil properties as a source of uncertainty affecting SAR-based soil moisture retrieval. The issues of radiometric accuracy of Sentinel-1, surface roughness parameter uncertainty and uncertain vegetation effects are much more pressing at field scale. The manuscript in it is current form is a review paper that does not adequately describe the state of the art in the field and is without any original research of little added value to the community.*

**Response A2**: The referee raises several points in this paragraph which we will come to in turn:

(a) "*The description of many elementary aspects of SAR remote sensing of soil moisture is incomplete*" - The paper does not set out to be a review of SAR remote sensing of soil moisture, but instead to focus on the contribution of soil properties and interactions with other factors. We will address these in our responses to the detailed comments.

(b) "*The prime focus on soil properties*" - The authors do not assert in this manuscript that soil properties are, by themselves, equally or more important than the other factors the referee lists. But we do present evidence of a correlation between soil properties and surface roughness for tilled soil. At the field scale the anthropogenic changes in surface roughness due to tilling are very significant and sudden and would contravene the assumptions of (for example) a change detection algorithm. Our work provides the basis for a future model to predict the change in surface roughness for arable fields over time, where one of the inputs would be the local soil texture. Such a model could be used to improve the performance of soil moisture retrieval at field scale for such land uses. Furthermore, we describe the change in dielectric constant of the soil when it freezes – this change is so significant that it leads to a major underestimate of the soil moisture. The magnitude of this error is strongly dependent on soil texture and could be estimated and corrected for, rather than simply masking the data. Our third point is that the end user or consumer of satellite-derived soil moisture is unlikely to be satisfied with data in the form of a relative index that may or may not be normalised to plant available water. They are likely to require a volumetric water content or soil moisture deficit for which they will need to use soil hydrological properties. The paper gives an estimate for the errors that could be introduced at this stage if an assumption is made about soil properties that is incorrect.

(c) "*The manuscript in it is current form is a review paper that does not adequately describe the state of the art in the field and is without any original research of little added value to the community*" – The research that the authors claim is original is as follows:

a. We report on the results of the implementation of the Park et al. (2017) model for a range of real example soils from the UK to establish a predicted relationship between the dielectric constant and the moisture content of the soil. The Park model was selected because of its capability to model the soil in the saturation regime. When dealing with surface soil moisture (as sensed by the satellite), saturation is very common after rainfall. However, Park et al. (2017) contained some minor errors and omissions, that were resolved by correspondence with the lead author to resolve these issues and arrive at a model implementation that would replicate the validated results. The next task was to take real soil properties data from the LANDIS database, and derive from it suitable input parameters for the model. Once achieved, the next problem was to invert the model so that we could determine the soil moisture given a measured value of dielectric constant, that might be obtained by analysis of SAR data. This was done by a trial and error method across a variety of soil types to determine the sensitivity of volumetric soil moisture estimation to the soil texture. The authors believe this is the first time such an analysis has been published.

b. The authors have reviewed agricultural journals and texts from as far back as 1970 to obtain data points from several sources representing measurements of soil surface roughness as a result tillage of operations followed by natural weathering. The analysis of Figure 2 has not been published before and provides a tentative basis for a future model to predict the magnitude of change and evolution over time of the surface roughness parameters within arable fields subject to conventional tillage.

c. The authors have added further data points from the literature (where the time evolution aspect was lacking) to examine the correlation between soil surface roughness after tilling and soil texture. The significance of this in the context of SAR soil moisture estimation is that there is now a basis for estimating the soil roughness from the soil texture, provided that a tillage operation is detected, and the type of operation assessed from knowledge of the crop cycle. We contend that this is original work and does add value.

**Detailed comments:**

*General: The abbreviation for soil moisture is not consistently used. Sometimes soil moisture is written out. Sometimes SM is used and on p3l13 mv is used.*

**Response A3**: The abbreviation for SM to represent soil moisture was defined at the first use on P1L19. Later, soil moisture was written out in full to aid readability. We accept that it is poor style to use both interchangeably, in the revised manuscript we will standardise to one or the other. $m_v$ on the other hand, is a commonly used abbreviation for the volumetric water content, which is a specific way of presenting soil moisture and the two terms are not equivalent.

*P2L17-33: The authors give an ambiguous description of the relationship between soil moisture and microwave (active/passive) observations, emission and backscattering. Soil moisture determines the dielectric constant and the dielectric constant is part of the refractive index that defines the specular reflection of microwave radiation, which is related to both the amount of microwave emission and backscattering.*

**Response A4**: The authors thank the referee for highlighting the potential confusion. This paragraph will be amended in the revised manuscript as follows:

"Soil moisture estimation by satellite remote sensing offers the potential for wide area coverage at minimal cost to the user. Thermal infrared (Hain et al., 2009) techniques exploit the thermal inertia of soil due to soil moisture, but other methods exploit the electromagnetic properties of the soil. The relative permittivity of the soil is a function of soil moisture and is a component of the soil's refractive index. By application of the Fresnel equations it can be shown that the reflectance of the soil is a function of refractive index, and therefore of soil moisture (Oh et al., 1992). By application of Kirchoff's Law relating emission to reflectance and emissivity, it follows that the emissivity of the soil is also a function of soil moisture. It is, therefore, feasible to retrieve soil moisture by both passive and active means. This may be attempted in the optical domain (Periasamy and Shanmugam, 2017), but all optical techniques are confounded by vegetation obscuring the soil, cloud cover, and certain lighting conditions. Passive microwave sensors are relatively unaffected by illumination or clouds but are characterised by a low spatial resolution (Drinkwater et al., 2009; Petropoulos et al., 2015; European Space Agency, 2017). Advantageously, L-band sensors are also minimally affected by vegetation, examples of these include SMOS (50km resolution) (Kerr et al., 2001) and the passive element of SMAP (40km resolution) (Entekhabi et al., 2010). Active radar sensors achieve higher spatial resolution, non-imaging examples include scatterometers (Bartalis et

al., 2007) and radar altimeters (Uebbing et al., 2017). Synthetic aperture radar (SAR) is an active microwave imaging technology with high spatial resolution (10-50m). Several L, C and X band 30 SAR satellites have been launched offering the spatial resolution and frequency of coverage that most closely addresses the requirement of near real-time field-scale moisture mapping. This study focuses on C-band because data from the European Space Agency's Sentinel-1 satellites is freely available, enabling regular derivation of soil moisture products at the required scale and resolution."

*P2L34: I agree with the authors that there are 'many unknown factors' that influence radar observations. In following sentence, however, the focus is on the soil properties, while from research it is well known vegetation, soil surface roughness and topography are much more important factors. In fact, much research (also cited later on in that manuscript) on how soil properties affect the relationship between soil moisture and the dielectric constant has been done.*

**Response A5**: Authors accept that vegetation, soil surface roughness and topography are significant factors and we discuss these in some detail at P3L25-26, and the sections that begin at P4L12, P5L4, and P6L16. Whilst the relationship between soil moisture and dielectric constant has been studied before, we demonstrate an apparent correlation between soil properties and soil surface roughness at the field scale, and a direct link between soil properties and the magnitude of the change in refractive index on freezing, for example. Much of the earlier work discussing the relationship between soil moisture and dielectric constant ignores organic matter content, and some papers even contradict each other, something we also refer to and attempt to quantify.

*P3L31: what do the authors mean by 'take account of incidence angle'? Do they mean that the backscatter is also determined by incidence angle? or that the derivation of the backscatter requires correction for the local incidence angle? Both are actually the case.*

**Response A6**: We mean the first of those. We propose to change the sentence as follows:
"The processing also has to take account of the dependence of the backscatter coefficient of soil on local incidence angle…"

*P3L31: 'validate' from this sentence it is not clear that it concerns here soil moisture validation. The jump from SAR processing to validation of soil moisture retrievals is for me too big. soil moisture validation.*

**Response A7**: Thank you - this is an omission; the authors propose to change this sentence to begin:
"To validate soil moisture estimates against ground measurements…"

*P4L7: R2*

**Response A8**: This omission is acknowledged with gratitude and will be corrected.

*P4L12: 'Vegetation . . .' this needs to be supported by a reference. Actually, I think that surface roughness the most important factor and not vegetation.*

**Response A9**: The relative importance of vegetation and surface soil roughness will depend on the type of vegetation cover and its density. The authors have cited Ulaby et al (1984) in the second sentence, and will add a reference to a recent paper by El Hajj et al. (2019)[2].In the revised manuscript, the opening sentences of this paragraph will be changed, as shown after the next two comments.
* * *
[2] El Hajj, M., Baghdadi, N., Bazzi, H. and Zribi, M. (2019) 'Penetration analysis of SAR signals in the C and L bands for wheat, maize, and grasslands', Remote Sensing, 11(1), pp. 22–24.

*P4L13: 'attenuating both the transmitted radar signal and the backscatter from the underlying soil'. Can the authors explain what the difference is between the 'transmitted radar signal' and 'the backscatter from the underlying soil'?*

**Response A10**: To clarify this, the opening sentences of this paragraph will be changed, as shown after the next comment.

*P4L17: This statement is not correct, and also depends on the incidence angle and vegetation type, see for instance Joseph et al. RSE 2010 or Mattia et al. TGRS 2003. Often the vegetation – soil scattering terms are responsible for a significant amount of soil moisture sensitivity beyond a LAI 0.5 m2 m-2.*

**Response A11**: The authors thank the referee for pointing us towards contrary evidence at Joseph et al. (2010)[3] and Mattia et al. (2003)[4]. However, in both cases the studies involved one crop type (corn and wheat respectively) and sensitivity to soil moisture was found to be highly dependent on incident angle, and on imaging modes (such as HH) not available in Sentinel-1. Authors accept that their statement was too general and propose addressing this in the revised manuscript. Along with addressing the previous point, the opening sentences of this paragraph will be revised to:

"The effect of vegetation on the sensitivity of C-band SAR to soil moisture is dependent on many factors such as the incidence angle, polarisation, crop type, and leaf area index (LAI). Joseph et al. (2010) and Mattia et al. (2003) reported soil moisture sensitivity in developed corn and wheat crops under particular imaging conditions. On the other hand, El Hajj et al. (2019), found that for an NDVI value greater than 0.7 in wheat and grassland there was no correlation between the radar backscattering coefficient at C-band (Sentinel-1) and soil moisture. Ulaby et al. (1984) reported that, for typical agricultural crops, when the Leaf Area Index (LAI) is above approximately 0.5, the vegetation contribution driving radar backscatter dominates over the soil properties (moisture and roughness). In the case of Sentinel-1 C-band SAR, where only VV and VH polarisations are available, and there are fixed incident angles, it is generally the case that penetration of incident radiation through dense vegetation is very small and any soil backscatter is also strongly attenuated."

*P4-5: Apart from the Water Cloud and MIMICS models, there are other models that are nowadays used more frequently by the community, e.g. the distorted born approximation by Lang and Sidhu (1983) was proposed by Kim et al. (TGRS 2013) for the SMAP radar product and the output of the Tor Vergata discrete scatterer model is used for SMOS soil moisture retrieval over forests.*

**Response A12**: Authors are aware of these other models being used and it is interesting to learn from the referee that they are being used more frequently than WCM and MIMICS. There are many recent papers that continue to report use of these models. Again, authors were not intending to present a comprehensive review of the modelling of vegetation canopies, the discussion is included for the sake of presenting an overview of the topic and is not the main focus of this manuscript. Authors will consider revising this paragraph to include more examples as suggested.
* * *
[3] Joseph, A.T., van der Velde, R., O'Neill, P.E., Lang, R. and Gish, T. (2010) 'Effects of corn on C- and L-band radar backscatter: A correction method for soil moisture retrieval', Remote Sensing of Environment, 114(11) Elsevier B.V., pp. 2417–2430.

[4] Mattia, F., Le Toan, T., Picard, G., Posa, F.I., D'Alessio, A., Notarnicola, C., Gatti, A.M., Rinaldi, M., Satalino, G. and Pasquariello, G. (2003) 'Multitemporal C-band radar measurements on wheat fields', *IEEE Transactions on Geoscience and Remote Sensing*, 41(7 PART I), pp. 1551–1560.

*P7L1: It would be better if the authors would state that the dielectric constant is defined as electric permittivity relative to the electric permittivity of free space.*

**Response A13**: That is correct, but to elaborate in this way would be to labour a point that is unnecessary to the argument. To avoid confusion authors propose removing the sentence in parentheses beginning on P7L1 and its associated reference.

*P7: Figure 1 suggests that there different models to simulate the bare soil backscatter, but that there is only the water cloud model to account for the effects of vegetation. This is a bit out of balance.*

**Response A14**: We do include the MIMICS model in that list and not just WCM. But the point is taken, and authors will add one or two further examples to the diagram in the revised manuscript.

**Reply to Referee #2 ([https://doi.org/10.5194/hess-2019-294-RC2](https://doi.org/10.5194/hess-2019-294-RC2), 2019)**

General Comments

*This is an interesting work investigating potential perturbing factors and "forgotten processes" of soil moisture retrieval from SAR data at C-band. This is useful because some of these limitations are not often presented in the existing literature, and because many applications derived from Copernicus Sentinel-1 data are emerging. From this point of view, the focus on the impact of soil temperature on permittivity is particularly interesting. The overview of recent results regarding soil roughness is also interesting for HESS readers. Comments on the limitation of the change detection approach are relevant. However, this work cannot be considered as a research paper. No new original result is presented. Category could be changed to "review paper". To reach this status, a more complete overview of the literature is needed (e.g. recent research on the use of interferometry to retrieve surface properties).*

*Recommendation: major revisions.*

**Response B1**: The authors thank the referee for their positive comments on the paper. We felt it necessary to set the context by citing some examples of current research in the field, but it was never the intention to present this as a comprehensive review. The original research is set out in Response A2(c), above.

**Detailed comments:**

*- P. 3, L. 18 ("Sigma0"): Yes, Sigma0 is useful. What about other quantities such as phase, coherence? The possible use of interferometric information to constrain soil moisture was investigated by a number of authors. E.g. Scott et al. ([https://doi.org/10.1038/s41598-017-05123-4](https://doi.org/10.1038/s41598-017-05123-4) ) and other works cited by Scott et al. This cannot be ignored.*

**Response B2**: We recognise that techniques based on phase and coherence may be used to estimate soil moisture, but such techniques are not as far advanced and there are as yet, no operational products available based on such techniques. Therefore, our manuscript focuses on the backscatter analysis. The authors have not looked at the impact of soil properties on the interferometric techniques. We would be open to consider a change to the title to reflect that the paper concerns backscatter techniques only. In the revised manuscript, authors will note the existence of interference techniques with a change to the text beginning at P2L18-23, and the addition of other references as follows:

"Two approaches for soil moisture retrieval from SAR include analysis of the SAR backscatter coefficient ($\sigma^0$), and interferometric techniques. The latter have been shown to provide a sensitivity to soil moisture due to variations in penetration depth (and so the depth range of the scattering centres) as described by De Zan et al. (2014)[5], Scott et al. (2017)[6], and Conde et al. (2018)[7]. However, these techniques involve more complex processing and are not yet as close to operational status as those based on backscatter, which we focus on in this paper. $\sigma^0$ is easily obtained or processed from online satellite data services and is a function of many variables associated with the viewing geometry and land surface, of which soil moisture is only one. ....."

*- P. 4, L. 16 (LAI): Is LAI the main factor or should vegetation water content or plant biomass be considered?*

**Response B3**: There are several plant biophysical parameters, such as LAI, biomass or vegetation water content, and remote sensing indices, such as NDVI, that could be used, to quantify vegetation. The authors have already proposed a change to this paragraph in response to a comment by Referee #1, the proposed revised text is shown as part of response A11, above,

*- P. 6, L. 9 (dew): What about intercepted rainwater?*

**Response B4**: Yes, that is a good point. A change to the text is proposed:
"        The presence of dew as a layer on the vegetation (and perhaps a temporary wetting of the soil surface) gives rise to another source of error in soil moisture estimation, similar to the problem of intercepted rainwater that remains on the leaves."

*- P. 9, L. 17 ("increase of NDVI"): This could be a matter of soil water holding capacity. E.g. Dewaele et al. (https://doi.org/10.5194/hess-21-4861-2017) found a relationship between maximum annual LAI values and SWHC for straw cereals.*

**Response B5**: Yes, SWHC is probably the key issue here, authors are grateful for the additional reference[8] to clarify this point. This will be included in a revised sentence as follows:
"However, it has been discovered (Farrar et al., 1994; Demattê et al., 2017) that the increase of NDVI over time following rainfall is dependent on soil texture and chemical composition, with clay soils and aluminium-rich soils showing the greatest rate of increase in NDVI. Similarly, Dewaele et al. (2017) found that maximum LAI observations were correlated with the maximum soil water holding capacity over France. It is currently unknown how useful these facts would be in the retrieval of soil 20 moisture from SAR."

*- P. 10, L. 11 ("for the others. . ."): Why ? mv values for soils 1-4 (less than 50%) sound more realistic to me than for soils 5-7 (above 50%). mv larger than 50% are usually observed for organic soils presenting a very large fraction of organic matter. Are soils 5-7 organic?*

[5] De Zan, F., Parizzi, A., Prats-Iraola, P. and López-Dekker, P. (2014) 'A SAR interferometric model for soil moisture', IEEE Transactions on Geoscience and Remote Sensing, 52(1), pp. 418–425.
[6] Scott, C.P., Lohman, R.B. and Jordan, T.E. (2017) 'InSAR constraints on soil moisture evolution after the March 2015 extreme precipitation event in Chile', Scientific Reports, 7(1) Springer US, pp. 1–9.
[7] Conde, V., Catalão, J. and Nico, G. (2018) 'Field observations of temporal variations of surface soil moisture: Comparison with InSAR sentinel-1 data', International Geoscience and Remote Sensing Symposium (IGARSS), 2018-July(Idl), pp. 6131–6134.
[8] Dewaele, H., Munier, S., Albergel, C., Planque, C., Laanaia, N., Carrer, D. and Calvet, J.C. (2017) 'Parameter optimisation for a better representation of drought by LSMs: Inverse modelling vs. sequential data assimilation', *Hydrology and Earth System Sciences*, 21(9), pp. 4861–4878.

**Response B6**: All the soils referred to are mineral soils. The referee observes that volumetric soil moisture values over 50% are rare in mineral soils, this may be true for most of the soil depth profile. However, at the surface, soils regularly reach their saturation point during and after rainfall events. Saturation values are related to porosity and not organic matter content. In the context of SAR observations from satellite, the SAR penetration depth may only be a few centimetres, so saturation conditions will be seen quite regularly. Therefore, it is important in our analysis and explains the choice of the Park model, as it does consider the dielectric properties of mineral soils in this regime.

*- P. 13, L. 1 (Fig. 3): Are they organic soils? mv values larger than 50% are quite uncommon for mineral soils. Why 10_C? Is a similar result obtained at 25_C for example?*

**Response B7**: All the soils referred to are mineral soils. We refer the referee to the response to their previous comment. We are interested in the regime of soil moisture above Field Capacity for the reasons given.

*- P. 4, L. 7: "2<0.22" ?*

**Response B8**: Thank you for spotting this error. It will be corrected to: "$R^2 < 0.22$".